# Instead of Rewriting Foreign Code for Machine Learning, Automatically Synthesize Fast Gradients

**William S. Moses**
MIT CSAIL
wmoses@mit.edu

**Valentin Churavy**
MIT CSAIL
vchuravy@mit.edu

## Abstract

Applying differentiable programming techniques and machine learning algorithms to foreign programs requires developers to either rewrite their code in a machine learning framework, or otherwise provide derivatives of the foreign code. This paper presents Enzyme[1], a high-performance automatic differentiation (AD) compiler plugin for the LLVM compiler framework capable of synthesizing gradients of statically analyzable programs expressed in the LLVM intermediate representation (IR). Enzyme synthesizes gradients for programs written in any language whose compiler targets LLVM IR including C, C++, Fortran, Julia, Rust, Swift, MLIR, etc., thereby providing native AD capabilities in these languages. Unlike traditional source-to-source and operator-overloading tools, Enzyme performs AD on optimized IR. On a machine-learning focused benchmark suite including Microsoft's ADBench, AD on optimized IR achieves a geometric mean speedup of 4.2 times over AD on IR before optimization allowing Enzyme to achieve state-of-the-art performance. Packaging Enzyme for PyTorch and TensorFlow provides convenient access to gradients of foreign code with state-of-the-art performance, enabling foreign code to be directly incorporated into existing machine learning workflows.

## 1 Introduction

Machine learning (ML) frameworks such as PyTorch [48] and TensorFlow [1] have become widespread as the primary workhorses of the modern ML community. Computing gradients necessary for algorithms such as backpropagation [32], Bayesian inference, uncertainty quantification [60], and probabilistic programming [16] requires all of the code being differentiated to be written in these frameworks. This is problematic for applying ML to new domains as existing tools like physics simulators [23, 10, 17, 18, 35], game engines, and climate models [58] are not written in the domain specific languages (DSL's) of ML frameworks. The rewriting required has been identified as the quintessential challenge of applying ML to scientific computing [4]. As stated by Rackauckas [50] "this is [the key challenge of scientific ML] because, if there is just one part of your loss function that isn't AD-compatible, then the whole network won't train."

To remedy this issue, the trend has been to either create new DSL's [35, 17, 43] that make the rewriting process easier or to add differentiation as a first-class construct in programming languages [44, 9, 61, 37]. This results in efficient gradients, but still requires rewriting in either the DSL or the differentiable programming language. Developers may want to use code foreign to a ML framework to either re-use existing tools or write loss functions in a language with an easier abstraction for their use case. While there exist reverse-mode automatic differentiation (AD) frameworks for various languages, using them automatically on foreign code for an ML framework is difficult as they still require rewriting and have limited support for cross-language AD and libraries[61, 33, 30, 36]. The two primary approaches to computing gradients are as follows.

```
float mag(const float* x);//Compute magnitude in O(N)
void norm(float* out, float* in) {
    // float res = mag(in); code motion optimization can move outside the loop
    for(int i=0; i<N; i++) { out[i] = in[i]/mag(in); }
}
```

```
// LICM, then AD, O(N)                      // AD, then LICM O(N^2)
void ∇norm(float* out, float* d_out,        void ∇norm(float* out, float* d_out,
           float* in, float* d_in) {                   float* in, float* d_in) {
  float res = mag(in);                        float res = mag(in);
  for (int i=0; i<N; i++) {                    for (int i=0; i<N; i++) {
    out[i] = in[i]/res;                          out[i] = in[i]/res;
  }                                            }
  float d_res = 0;                            for (int i=0; i<N; i++) {
  for (int i=0; i<N; i++) {                     float d_res = -in[i]*in[i]/res \
    d_res += -in[i]*in[i]/res * d_out[i];                            * d_out[i];
    d_in[i] += d_out[i]/res;                    d_in[i] += d_out[i]/res;
  }                                             ∇mag(in, d_in, d_res);
  ∇mag(in, d_in, d_res);                      }
}                                           }
```

Figure 1: *Top:* An $O(N^2)$ function norm which normalizes a vector. Running loop-invariant-code-motion (LICM) [45, Sec. 13.2] moves the $O(N)$ call to mag outside the loop, reducing norm's runtime to $O(N)$. *Left:* An $O(N)$ ∇norm resulting from running LICM before AD. Both mag and its adjoint ∇mag are outside the loop. *Right:* An $O(N^2)$ ∇norm resulting from running LICM after AD. ∇mag remains inside the loop as it uses a value computed inside the loop, making LICM illegal.

*Operator-overloading* computes derivatives by providing differentiable versions of existing language constructs. Examples include Adept [33]/ADOL-C [27], C++ libraries providing differentiable types; and JAX [9]/Autograd [44], Python libraries providing derivatives of NumPy-style functions. These approaches, however, require rewriting programs to use differentiable operators in place of standard language utilities. This prevents differentiation of many libraries and code in other languages.

*Source-rewriting* [26] analyzes the source code of programs and emits source code defining the gradient. Examples of tools include Tapenade [30, 47] for C and Fortran; ADIC [46] for C and C++; and Zygote [36, 38, 37] for Julia. Users must provide all code being differentiated to the tool ahead-of-time and must write programs in a specific subset of the language. This makes source-rewriting hard to use with header-only libraries and impossible to use with precompiled libraries.

Both operator-overloading and source-rewriting AD systems differentiate programs before optimization. Performing AD on unoptimized programs, however, may result in complicated gradients that cannot be simplified by future optimization. As an example, the gradient of norm in Figure 1 runs in $O(N)$ if optimization is run before AD and $O(N^2)$ if optimization is run after AD.

Traditional AD systems have not operated on optimized intermediate representation (IR) as doing so requires either re-implementing all of the optimizations or working at a low-level after which optimization has already been performed. Conventional wisdom says that producing efficient gradients for low-level IR is difficult as it lacks high-level information many tools rely upon: "AD is more effective in high-level compiled languages (e.g. Julia, Swift, Rust, Nim) than traditional ones such as C/C++, Fortran and LLVM IR [...]" – Innes [36]. This paper challenges that wisdom by creating an efficient AD tool for LLVM [41], a low-level IR and set of optimizations used by many compilers.

This paper presents Enzyme, an efficient cross-platform compiler plugin for automatic differentiation that operates on LLVM IR [41] and makes the following contributions:

- Enzyme, a compiler plugin for LLVM that can synthesize fast gradients of statically analyzable LLVM IR, including IR generated by compiler frontends for C, C++, Fortran, Rust, Swift, etc.
- PyTorch-Enzyme/TensorFlow-Enzyme, a foreign-function interface that allows machine learning researchers to use foreign code written in LLVM-compiled languages in PyTorch and TensorFlow.
- Enzyme.jl, a Julia package that uses Enzyme to synthesize gradients of code written in a dynamic high-level language using only low-level information.
- Multisource AD and static library support by leveraging link-time optimization (LTO) [41, 39].
- A study demonstrating that running AD after optimization results in significant performance gains on a standard machine learning benchmark suite [57] and achieves state-of-the-art performance.

```
void f(void* dst, void* src) { memcpy(dst, src, 8); }
// Gradient memcpy for double inputs          // Gradient memcpy for float inputs
void grad_f(double* dst, double* ddst,        void grad_f(float* dst, float* ddst,
            double* src, double* dsrc) {                  float* src, float* dsrc) {
  // Forward pass                               // Forward pass
  memcpy(dst, src, 8);                          memcpy(dst, src, 8);
  // Reverse pass                               // Reverse pass
  dsrc[0] += ddst[0];                           dsrc[0] += ddst[0];
  ddst[0] = 0;                                  ddst[0] = 0;
                                                dsrc[1] += ddst[1];
                                                ddst[1] = 0;

}                                             }
```

Figure 2: *Top:* Call to `memcpy` for an unknown 8-byte object. *Left:* Gradient for a `memcpy` of 8 bytes of double data. *Right:* Gradient for a `memcpy` of 8 bytes of float data.

**Related work**   Clad is a plugin to the Clang compiler that implements forward mode automatic differentiation on a subset of C/C++ with reverse mode in development [59]. Chen et al. [11] present an end-to-end differentiable model for protein structure prediction. DiffTaichi [35] implements a differentiable DSL for physics and robotics simulation. de Avila Belbute-Peres et al. [17] also provide a differentiable physics framework. Halide is a differentiable DSL for image processing [43]. Swift implements first class automatic differentiation [61]. Elliott [21] present a compiler plugin to provide differentiable programming in Haskell. Enzyme differs from the related work by running on generic low-level IR and post-optimization. This gives Enzyme several performance and compatibility benefits that don't exist in current systems.

## 2   Design

Enzyme is composed of three stages: *type analysis* determines the underlying types of values, *activity analysis* determines what instructions and values can impact the gradient calculation, and *synthesis* creates the necessary functions to compute the gradient. A core design goal of Enzyme is to operate upon optimized IR. As seen in Figure 1 this can result in significant benefits such as simpler and more optimized gradients, though it requires working on a low-level representation. Gradients synthesized by Enzyme contain two parts: a *forward pass* that mirrors the original code and a *reverse pass* that computes the gradient by inverting the instructions in the forward pass. Inverted instructions in the reverse pass are known as *adjoint*s. For all differentiable instructions in LLVM, Enzyme defines an adjoint to describe how gradients propagate through each instruction.

**Type Analysis**   One challenge of performing AD on LLVM IR (and even C/C++) is that LLVM types do not necessarily represent the type of the underlying data. For example, the `memcpy` function copies data between generic pointers without types (`void*`). Creating a correct gradient for `memcpy`, however, requires knowing the type of the memory being copied. As shown in Figure 2, copying 8 bytes of double data requires performing one double (8-byte) addition in the reverse pass, whereas copying 8 bytes of float data requires two float (4-byte) additions. These operations are incompatible, resulting in an incorrect gradient if the wrong one is used.

Since Enzyme works on a low-level representation, Enzyme must use a new interprocedural fixed-point analysis rather than relying on types prescribed by the language. Every value in a function is given a *type tree* that describes the known type at any given byte offset in the value. If the type at a particular offset is a pointer type, we have a new type tree that represents the types inside that offset. An example type tree is shown in Figure 3.

Type analysis initializes the type trees of all values to empty and uses type-based alias analysis (TBAA) metadata to initialize the type trees of loads, stores, and `memcpy` operations. TBAA allows us to make assumptions about the underlying type because of strict aliasing [19, 12]. For every kind of instruction, Enzyme implements a type propagation rule that specifies how types flow through the instruction. As an example, if the result of a load is known to be type T, then the pointer loaded must be a pointer to T at offset 0. Type analysis then runs all of the type propagation rules until a fixed point is reached. This is an application of abstract interpretation [15].

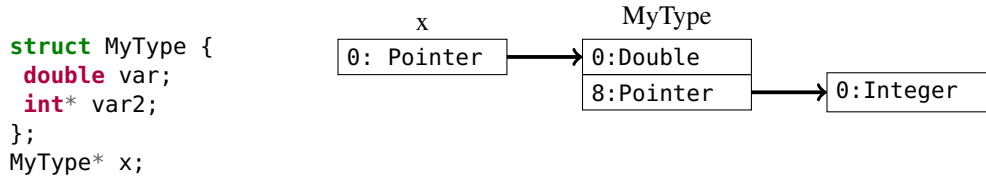

```
struct MyType {
  double var;
  int* var2;
};
MyType* x;
```

Figure 3: An example TypeTree used by Type Analysis. The variable x (declared on the left) is a pointer type, which points to a struct MyType, which contains a double at byte 0, and then a pointer at byte 8. That nested pointer points to an integer.

```
double sum(double* x) {
  double total = 0;
  for(int i=0; i<10; i++)
    total += read() * x[i];
  return total;
}

void grad_sum(double* x,
              double* d_x) {
  double* read_cache = malloc(10*8);
  for(int i=0; i<10; i++)
    readCache[i] = read();
  // reverse
  for(int i=10-1; i>=0; i--)
    d_x[i] += read_cache[i];
  }
  free(read_cache);
}
```

```
double g(double* x) { return *x * *x; }
void f(double* x) { *x = g(x); }

{/*return val*/double,/*cache*/double}
augmented_g(double* x) {
  return {x[0]*x[0], x[0]};
}

void grad_g(double* x, double* d_x,
            double d_ret, double cache) {
  d_x[0] += 2 * cache * d_ret;
}

void grad_f(double* x, double* d_x) {
  {call, cache} = augmented_g(x);
  *x = call;
  double d_ret = *d_x;
  *d_x = 0;
  grad_g(x, d_x, d_ret, cache);
}
```

Figure 4: *Left:* Caching the result of read for the reverse pass. *Right:* Creating an augmented forward pass for a function to ensure requisite values are cached for the reverse.

Sometimes Type Analysis cannot deduce all the necessary information statically (e.g. if bithacks to modify a floating-point). Rather than produce incorrect code, Enzyme will emit a compile-time error if it is unable to perform an analysis needed by AD. This enables programmers to provide this information to the compiler in the form of additional attributes, a custom derivative, or other means.

**Activity Analysis** Activity analysis determines what instructions could impact the gradient computation and is common in automatic differentiation systems to avoid performing unnecessary adjoints [56, 8]. Enzyme also uses activity analysis to avoid taking gradients of "undifferentiable" instructions such as the cpuid instruction. An instruction is active if and only if it can propagate a differential value to its return or another memory location. For example, a function that counts the length of a an active input array would not be active. In our implementation of activity analysis, we leverage LLVM's alias analysis [3, Ch. 12] and type analysis to help prove that instructions are inactive. As an example, any read-only function that returns an integer must be inactive since it cannot propagate differential values through the return or any memory location. This is true because the differential value of any integer value must be zero and while the instruction can read active memory it cannot propagate it anywhere.

**Shadow Memory** Shadow memory is common in AD systems as a way to store gradients of values. Consider the gradient of sum in the left of Figure 4. The gradient function grad_sum takes in both x as an argument as well as the shadow d_x, where it will store the result. Enzyme's scheme is designed to be amenable to optimizations in LLVM while maintaining sufficient flexibility to represent arbitrary programs. For every active value in the forward pass, Enzyme creates and zeros a shadow version of that value. Similarly, any data structures (including function arguments) need to be duplicated. For any data structures computed inside the function being differentiated, Enzyme will create a shadow data structure automatically. This involves duplicating any memory instructions such as malloc, new, and stores of pointers, with equivalent shadow memory operations. Finally, Enzyme delays all deallocations until the memory is not needed by the gradient calculation. Shadow memory is used to

```
define double @relu3(double %x)          reverse_if.end:
entry:                                     ; adjoint of return
  ; Shadow values for reverse              store %d_res = 1.0
  ; alloca %d_x = 0.0                       ; adjoint of %res phi node
  ; alloca %d_call = 0.0                   %d_call += if %x > 0, (load %d_res), else 0
  ; alloca %d_result = 0.0                 store %d_res = 0.0
  br (%x > 0), if.true, if.end             br %cmp, %reverse_if.true, %reverse_entry
if.true:                                 reverse_if.true:
  %call = @pow(%x, 3)                       ; adjoint of %call
  br cond.end                              %df = 3 * @pow(%x, 2)
if.end:                                    %d_x += %df * (load %d_call)
  %res = phi[%call, if.true],              store %d_call = 0.0
  ↪  [0, entry]                            br %reverse_entry
  ret %res                               reverse_entry:
                                           %0 = load %d_x
                                           ret %0
```

Figure 5: Example gradient synthesis for `relu(pow(x,3))`. The left hand side shows the LLVM IR for the original computation. In the comments on the left we show the shadow allocations of active variables that would be added to the forward pass. The right hand side shows the reverse pass that Enzyme would generate. The full synthesized gradient function would combine these (with shadow allocations added), replacing the return in `if.end` with a branch to `reverse_if.end`.

compute the adjoint of instructions like `load` in the reverse pass, which propagates the gradient of the load to the shadow of the pointer operand. Given shadow versions of all arguments and active globals, the shadow version of any value can be computed by duplicating the instruction that created the original value, replacing operands with their shadow. For calls to functions, we return the shadow pointer along with the original pointer.

**Synthesis** Given the results of type and activity analysis, Enzyme can now perform synthesis, the creation of the gradient function. Enzyme initializes all the shadow values as described above. For every basic block BB in the original program, Enzyme creates a corresponding reverse block `reverse_BB`. Enzyme then emits the adjoint of all instructions from BB into `reverse_BB` in reverse order. Enzyme then branches to the reverse of BB's predecessor, returning if BB was the entry block. Finally, Enzyme replaces any return instruction in the forward pass with a branch to its reverse block. An example of this procedure is shown in Figure 5.

**Cache** Computing adjoints of certain instructions requires values computed in the forward pass. By default, Enzyme will attempt to recompute these in the reverse pass. However, it may be impossible or less efficient to recompute certain instructions. The question of whether and how to cache is known as the well-studied "checkpointing" problem in the literature [25, 40]. Checkpointing in Enzyme adds additional complexity with the inclusion of potentially-aliasing memory, a cost model for LLVM instructions (many of which are cost-free), and the impact of checkpointing on future optimization.

Consider the calls to `read` on the left of Figure 4, which cannot be recomputed. Enzyme provides a cache (often referred to as a tape in other AD systems) that provides forward-pass values to the reverse pass. In this example, Enzyme allocates memory (in this case an array of 10 doubles) to store the values needed by the reverse pass. If Enzyme can statically bound the number of values needing to be cached (e.g. a loop of fixed size), it will perform a single allocation to cache that instruction. If not, Enzyme will dynamically reallocate memory. For function calls, Enzyme may need to augment a call in the forward pass as shown in the right of Figure 4 to save values needed to compute the adjoint of the call.

To maximize performance, it is often desirable to reduce the number of values cached and Enzyme contains optimizations to reduce the number of values that need caching. Enzyme greatly benefits from LLVM's alias analysis and function attributes by proving that it is legal to recompute certain instructions. Enzyme also runs a differential-use analysis to determine which values are not necessary for computing the gradient and avoids caching them. This analysis is sometimes referred to as "to be recorded analysis" in other systems [31]. Additionally, if Enzyme already cached an equivalent value (e.g. a load to the same location which couldn't have since been written to), Enzyme simply reuses the existing cache for that value. Finally, if a cached value $A$ is only used to recompute a single value

```
__attribute__((
  enzyme("augment", augment_f),
  enzyme("gradient", gradient_f)
))
double f(double in);
double func(double* x, double* y) {
    return f(*x) + f(*y);
}
```

```
double dfunc(double* x, double *d_x,
             double* y) {
  __enzyme_autodiff(func,
      // The variable x is active
      enzyme_dup, x, d_x,
      // The variable y is constant
      enzyme_const, y);
}
```

Figure 6: *Left:* Specifying a custom forward and reverse pass for `f`. *Right:* Creating a gradient for func with $x$ as an active variable and $y$ as a constant.

$B$ in the reverse pass, Enzyme will choose to cache the value $B$ instead of the value $A$, minimizing the amount of work in the reverse pass.

**Function Calls** It is desirable to compute both the forward and reverse pass in the same function. This allows for optimization between the forward and reverse pass, and can reduce memory usage. Enzyme detects whether it is legal to move the forward pass instructions of a function into the adjoint computation. If so, the forward pass call is erased and the combined function is used as the adjoint.

An ***indirect function call*** is a call to an anonymous function pointer which is not known at compile time. Like all other active pointers in a function, there exists a shadow version of the function pointer being called. Whenever a function pointer is used outside of a static call, we create a new global variable containing a pair of functions, namely the augmented forward and reverse pass. This global is then used as the shadow pointer for the original function. Thus, whenever Enzyme needs to perform an adjoint of an active indirect function call, it extracts the augmented forward and gradient functions from the shadow of the indirect callee, then uses those functions in the adjoint. Like the rest of shadow memory, this is handled automatically by Enzyme for all objects created inside functions being differentiated. If you want Enzyme to differentiate a function with a virtual C++ class as an argument, however, you need to pass in a modified virtual method table in the shadow that conforms with Enzyme's calling convention.

**Limitations** Enzyme needs access to the IR for any function being differentiated to create adjoints. This prevents Enzyme from differentiating functions loaded or created at runtime like a shared library or self-modifying code. Enzyme also must be able to deduce the types of active memory operations and phi nodes. Practically, this means enabling TBAA for your language and limiting yourself to programs with statically-analyzable types (no unions of differing types nor copies of undefined memory). Enzyme presently does not implement adjoints of exception-handling instructions so exceptions should be disabled (e.g. with `-fno-exceptions` for a C++ compiler).

## 3   Usage

Enzyme is designed to simplify both importing foreign code into machine-learning workflows and providing native AD for LLVM-based languages. Enzyme is implemented as an LLVM compiler plug-in, allowing it to be easily used in existing tools without need to build and maintain custom forks of LLVM, PyTorch, or TensorFlow.

**Static Languages** Using gradients inside LLVM-based languages simply requires calling an external `__enzyme_autodiff` function as shown on the right in Figure 6. For added control, users may specify whether a variable is active by including either an Enzyme-specific variable or metadata as part of the function call. Enzyme requires the IR for all functions it may need to differentiate to be available when the pass is run. For single-source programs, all the IR is simply available. Codebases with multiple source files or those using external libraries require an additional step. Enzyme makes use of Link-Time Optimization (LTO) [39, 41], a compiler technique for whole-program optimization that preserves IR from all source files until link time where a final set of interprocedural optimizations may run. To use Enzyme on multi-source codebases, a user enables LTO and runs Enzyme on the merged IR for all the sources. Static libraries are handled by compiling them with the `-fembed-bitcode` command that ensures that bitcode is included in the library as well. This allows Enzyme to perform AD on a program linking against a static library, by extracting the bitcode in the static library and then running Enzyme on the original program with the IR of the static library.

```
function f(x)
   sum = zero(x)
   for i = 1:10^7
      sum += x^i / i
   end
   return sum
end
```

| Tool       | Runtime (s) |
|------------|-------------|
| Enzyme.jl  | 0.810       |
| Zygote.jl  | 24.638      |
| AutoGrad.jl| 609.256     |

```
using Zygote, Enzyme

Zygote.@adjoint f(x),
   Enzyme.pullback(f, x)

Zygote.gradient(f, 0.5)
```

Figure 7: *Left:* A simple scalar function computing a Taylor expansion. *Center:* The runtime of the gradient as computed by Enzyme.jl and two common Julia AD frameworks. *Right:* How Enzyme can be embedded in existing AD frameworks to use Enzyme's efficient implementation of scalars.

Programmers can use custom forward and backward passes in Enzyme by specifying them as metadata on the function to be differentiated, even if the definition of that function is not available during AD. In a separate Clang C/C++ frontend extension, we allow users to specify this directly with function attributes as on the left in Figure 6. Internally, one can also specify the type propagation, activity analysis, and adjoint rules for custom foreign functions. To minimize the amount of work for users, we provide these rules for common functions in the C/C++ standard and math libraries.

**Dynamic Languages** Dynamic languages such as Julia require more consideration. Julia uses LLVM to perform native code generation for functions as a Just-In-Time compiler. The IR for all code needed by Enzyme is not immediately available since Julia's execution engine uses caching aggressively. We use the infrastructure developed for Julia's GPU code generator [5, 6] to collect all the function definitions reachable by the function to be differentiated. Julia implements its own version of common math functions like `sin` with custom implementations that are not amenable to type analysis, or resolves them to indirect function calls through opaque pointers into `libm`. `Enzyme.jl` uses `Cassette.jl` [53] to replace these functions calls with LLVM intrinsics. The Enzyme plugin is loaded and the Enzyme pass directly executed over the collected IR.

Zygote [36, 38, 37] is a popular automatic-differentiation framework for Julia used in probabilistic programming [24] and scientific machine learning [51]. Zygote performs source-to-source AD on high-level Julia code with optimizations for matrix programs. As shown in Figure 7, however, it can perform poorly on scalar programs. By embedding Enzyme inside Zygote as shown in the right of Figure 7, Julia is able to perform AD with both high-level knowledge and low-level optimizations. By utilizing embedded bitcode, `Enzyme.jl` provides the ability to take derivatives of foreign functions.

**ML Frameworks** Having demonstrated the ability to synthesize gradients of functions in a variety of languages compiled by LLVM, we will demonstrate how to leverage this ability to embed foreign code into a machine learning framework. After specifying the desired gradient by calling `__enzyme_autodiff` as shown in Figure 8, users can follow the tutorials for creating a custom operator in PyTorch [14] or TensorFlow [13] and compiling the custom operator with Enzyme as described above. To simplify this workflow for machine learning researchers, we also created a simple package for PyTorch and TensorFlow in Figure 8 that exposes this functionality in Python without needing to compile a custom operator.

## 4   Evaluation

We evaluate the Enzyme approach by measuring the run time of seven benchmarks: the three reverse-mode automatic differentiation benchmarks from Microsoft's machine learning-focused ADBench suite [57], and four additional tests that are technically interesting or represent potential uses of Enzyme in practice. The ADBench suite includes bundle analysis (BA), a long short term memory model (LSTM), and a gaussian mixture model (GMM). We also differentiate two integrators (Euler, RK4) from the Odeint header-only ODE solver library [2]; a simple Fast Fourier Transform (FFT); and a finite difference discretized simulation of the 2-dimensional Brusselator system (Bruss) [22, 62].

The two integrators test indirect function calls, complicated C++ headers, and foreign ODE solvers. The FFT test demonstrates AD of recursive functions. The Brusselator test demonstrates the utility in adjoint sensitivity analysis for ordinary differential equations, a widely applicable method with applications to PDE-constrained optimization [7, 42], control theory [49], and scientific machine learning like neural ODEs [51, 11].

```
void f(float* inp, size_t n, float* out); // Input tensor + size, and output tensor
void diffef(float* inp, float* d_inp, size_t n, float* d_out) {
  // enzyme_dupnoneed specifies not recomputing the output
  __enzyme_autodiff(f, enzyme_dup, inp, d_inp, n, enzyme_dupnoneed, (float*)0, d_out);
}
```

```
import torch                              import tensorflow as tf
from torch_enzyme import enzyme           from tf_enzyme import enzyme
# Create some initial tensor
inp = ...                                 inp = tf.Variable(...)
# Apply foreign function to tensor        # Use external C code as a regular TF op
out = enzyme("test.c", "f").apply(inp)    out = enzyme(inp, filename="test.c",
# Derive gradient                                      function="f")
out.backward()                            # Results is a TF tensor
print(inp.grad)                           out = tf.sigmoid(out)
```

Figure 8: ***Top:*** Sample glue code for using Enzyme to produce a custom operator for an ML framework. ***Left & Right:*** Sample code of using Enzyme to provide gradients of foreign code in PyTorch and TensorFlow, respectively.

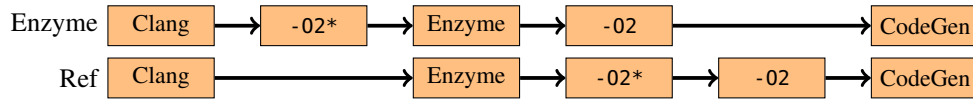

Figure 9: The pipelines Enzyme and Ref, which run optimizations before and after AD, respectively. The goal of running optimizations prior to AD is to reduce work and simplify the code. The first round of optimizations (-O2*) disables scheduling passes such as vectorization or unrolling that make heuristic decisions based on the current code size and machine attributes. Scheduling optimizations are included in the second round of optimizations (-O2) when the entire code (including gradient) is available.

We ran our experiments on a "quiesced" AWS c4.8xlarge instance with hyperthreading and Turbo Boost disabled. For all benchmarks, we took the geometric mean across all inputs. We ran all 92 inputs from ADBench, removing the 21 inputs where Adept exhausted system memory or a tool ran in under 0.01 seconds. For the integrator and FFT tests, we ran a total of 36 different inputs, with the number iterations or the input size increasing exponentially. For Bruss, we ran a total of 10 trials.

To evaluate the effectiveness of AD on optimized IR, we construct two pipelines shown in Figure 9. The Enzyme pipeline consists of running optimizations before Enzyme AD, followed by a second round of optimizations. The Reference (Ref) pipeline is identical to the Enzyme pipeline, except that AD is performed before the first round of optimization. This allows us to effectively evaluate

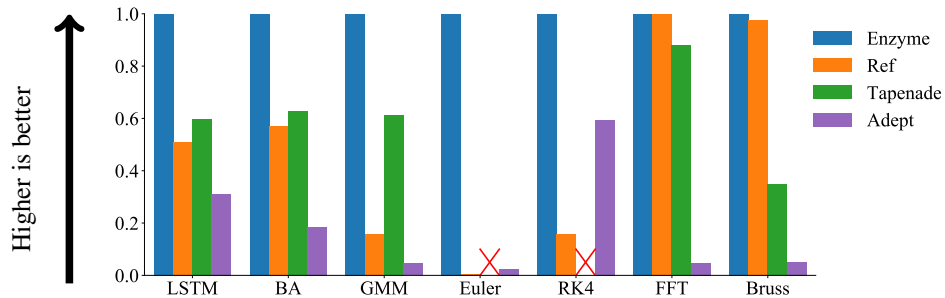

Figure 10: Relative speedup of different AD systems on the benchmark suite, higher is better. A red X is used to denote a system not being compatible with the benchmark (Tapenade only supports C and not C++ programs). For each benchmark, we take the geometric mean of the run time for all test cases, normalizing to the victor. A value of 1.0 denotes the fastest AD system tested for that benchmark, whereas a value of 0.5 denotes that an AD system produced a gradient which took twice as long.

|        | Enzyme | Ref    | Tapenade | Adept |
|-------:|:------:|:------:|:--------:|:-----:|
| LSTM   | **2.408** | 4.727 | 4.033 | 7.722 |
| BA     | **0.256** | 0.450 | 0.408 | 1.380 |
| GMM    | **0.076** | 0.480 | 0.125 | 1.677 |
| Euler  | **0.165** | 29.453 | N/A | 6.954 |
| RK4    | **3.936** | 25.015 | N/A | 6.632 |
| FFT    | **0.122** | **0.122** | 0.139 | 2.632 |
| Bruss  | **0.180** | **0.184** | 0.513 | 3.546 |

Table 1: Geometric mean runtime of benchmark suite in seconds. Tapenade compiles only C and not C++. N/A denotes a system incompatible with the benchmark (Tapenade only supports C and not C++ programs).

the importance of optimization on AD without considering additional confounding factors (such as differing tape implementations) between Enzyme and existing source AD systems. Taking the geometric mean across all benchmarks and inputs, Enzyme outperforms Ref by a factor of 4.2.

We also compare against the two fastest C/C++ AD tools evaluated in ADBench, Tapenade and Adept[2]. These results are presented in Figure 10. Enzyme demonstrates state-of-the-art performance in all benchmarks. Enzyme's advantage in the BA, LSTM, Euler, and RK4 tests appears to stem from running optimizations before AD. Enzyme uses a different tape structure than Tapenade (using a recursive set of allocations rather than a stack), which explains their differences on the GMM and Bruss benchmarks. Enzyme does not need to store as much on its tape as Adept (such as not needing to store which statements were executed), explaining Enzyme's superior performance on FFT and Bruss.

## 5 Conclusion

Enzyme demonstrates the feasibility of performing efficient AD on low-level programs, opening up the door for language-independent AD and AD after optimization. This transforms the existing workflow machine learning researchers use to bring ML to foreign code. Instead of rewriting foreign code for machine learning, they can automatically synthesize fast gradients! This allows researchers to apply ML to a vast array of new use cases without the substantial effort of a rewrite or new DSL.

Building Enzyme as part of the LLVM compiler creates many avenues for future research. Exploring new AD-specific optimizations in LLVM may yield additional performance benefits. One could use LLVM's existing GPU or parallel code generators on programs generated by Enzyme [28, 29]. Enzyme could be extended to differentiate GPU and CPU-parallel programs by using existing representations for these programs in LLVM [54, 34, 55, 20]. Enzyme could also be extended to support forward-mode AD, mixed-mode AD [52], and the checkpointing problem beyond a simple heuristic. Fine-tuning the location of Enzyme in LLVM's optimization pass pipeline remains an open question. Enzyme opens up opportunities for cross-language AD. There are also opportunities to use Enzyme to port various physics engines and other codebases to ML frameworks.

## Acknowledgments and Disclosure of Funding

Special thanks to Tim Kaler of MIT for introducing the topic of AD to the authors. Tim's work on AD inspired Enzyme and Tim was a master bugfinder for an early prototype of Enzyme, contributing many test cases and patches. Thanks to Charles Leiserson of MIT for working with us to find a good title and suggesting edits. Thanks to Laurent Hascoet of Inria for caching discussions and helping run Tapenade on various codes. Thanks to Yingbo Ma and Chris Rackauckas of MIT for their help in understanding scientific ML and its relationship to AD. Finally, the authors thank James Bradbury (Google), Alex Chernyakhovsky (Google), Hal Finkel (ANL), Paul Hovland (ANL), Jan Hueckelheim (ANL), Mike Innes (Julia Computing), TB Schardl (MIT), Lizhou Sha (University

of Wisconsin–Madison), Yo Shavit (Harvard), Dhash Shrivathsa (Radix Labs), Nalini Singh (MIT), Miguel Young de la Sota (Google), and Alex Zinenko (Google) for their invaluable feedback and advice.

William S. Moses was supported in part by a DOE Computational Sciences Graduate Fellowship DE-SC0019323. Valentin Churavy was supported in part by the Defense Advanced Research Projects Agency (DARPA) under Agreement No. HR0011-20-9-0016, and in part by NSF Grant OAC-1835443. This research was supported in part by Los Alamos National Laboratories grant 531711. Research was sponsored in part by the United States Air Force Research Laboratory and was accomplished under Cooperative Agreement Number FA8750-19-2-1000. The views and conclusions contained in this document are those of the authors and should not be interpreted as representing the official policies, either expressed or implied, of the United States Air Force or the U.S. Government. The U.S. Government is authorized to reproduce and distribute reprints for Government purposes notwithstanding any copyright notation herein.

## Broader Impact

Enzyme reduces the amount of work necessary to apply ML to new domains. This has a generally positive impact as it reduces the workload necessary by researchers to use ML. It could be negative, however, for those whose job manually rewrites existing code for ML frameworks. Similarly, this added accessibility advances various scientific problem domains with all the positives and negatives that come with it. Enzyme also provides generally positive impact by helping bridge the gap between the ML and the scientific computing communities through allowing them to share tools and more easily interoperate. As an example, Enzyme may allow for improved policy design for climate change via projected-gradient-descent on a climate simulator.

## Footnotes

[1]Code and documentation at `https://github.com/wsmoses/Enzyme` and `https://enzyme.mit.edu`.

[2]For ADBench benchmarks, Tapenade and Adept had their ADBench implementations were evaluated directly. Tapenade and Adept versions of benchmarks outside ADBench were generated via Tapenade's web interface or replacing programs with Adepts differentiable types, using Vector and Matrix extensions where relevant. All benchmarks are available on Github.

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
