[Supplementary Material]

# Instead of Rewriting Foreign Code for Machine Learning, Automatically Synthesize Fast Gradients

## 1 Experiment Details

In this appendix, we demonstrate how a reviewer can reproduce our experimental results. Please note that all data and code provided is intended soley for use in this review and the authors retain all copyrights to relevant pieces of code.

First, create an AWS c4.8xlarge instance, running Ubuntu 18.04 with a disk of size 128GB. This process will take around an hour.

```
$ # Log into AWS Instance
$ ssh -i mykey.pem ubuntu@my.instance.amazonaws.com
$ # Download Enzmye test code and data
$ wget https://anonymous-data.s3.amazonaws.com/enzyme.tar.gz
$ # Decompress Enzyme test code and data
$ tar -xzf enzyme.tar.gz
$ # Compile LLVM 8, Enzyme and run all the benchmarks
$ # This is the command that will take the hour to run
$ # We encourage going for a walk or getting tea
$ ./enzyme/benchmarks/cleantest.sh
```

To get the data from the Enzyme pipeline (running optimization before AD):

```
$ cd ~/enzyme/benchmarks
$ ./getdata.sh
```

To get the data from the Ref pipeline (running after before AD):

```
$ cd ~/enzyme/benchmarks
$ ./getdataafter.sh
```

An output of 88888888 denotes an out of memory error (OOM) on a test.

Names of the results printed out correspond to those in the graph/table with the following exceptions: The ode-const test corresponds to Euler, the ode test corresponds to RK4, and the ode-real test corresponds to Bruss.

To reproduce our results, remove the tests where Adept OOM's in GMM and the single case in BA that has a runtime under 0.00001s. Then for each benchmark, take the geomean of all test cases.

Note that you cannot run the `cleantest.sh` script multiple times as running it once will result in the Reference pipeline data overwriting the Enzyme data. If you do want to run this multiple times, uncompresss `enzyme.tar.gz` into a clean directory and run the script in the clean directory.