[Reviews · NeurIPS 2020]

Review 1

Summary and Contributions: The key contribution of this paper is a system called Enzyme automatic generation of code for differentiation. While this idea has seen a lot of interest over the last few years, the novelty in this particular proposal is the fact that the generation of code is for the LLVM IR. The main argument for this approach made in the paper is that such generation of code is post-optimization, though intuitively I find it difficult to understand why this is an important feature: it is not obvious that it is a better idea to generate code for computing a derivative before optimization (and to then optimize the generated code using normal compiler tools) than it is to generate code after optimization. That said, the authors do show experimentally that the generate-after-optimization approach is far superior (the generate-before-optimiztation approach is tested as the “ref” option in the paper, and it is often twice as slow as Enzyme). While this non-intuitive result is impressive, I feel that the main argument for the approach is that by doing this at the LLVM level, it is possible to plug the auto-diff software into any LLVM language, there’s no more need to work on language-specific auto-diff capabilities. Technically, the paper begins by describing the design of Enzyme. Enzyme runs in three stages. First is type analysis, where for every type of instruction, Enzyme has a rule describing how types propagate through the instruction, and it runs all of these rules until a fixed point is reached. Next is activity analysis, where Enzyme figures out which instructions could possibly affect the gradient computation, using LLVM’s alias and type analysis facilities to prove that an instruction is inactive. Finally, Enzyme performs synthesis, where each basic block is converted to a reverse block, and all gradients are stored in shadow memory. One challenge is the fact that in order to compute gradients, the result of the forward pass must be retained. The paper considers checkpointing of such values, and the potential of recomputing rather than storing values. Te paper also considers issues with static and dynamic languages, such as Julia. Experiments consider a small suite of seven benchmark computations, and Enzyme is compared with Tapenade and Adept. Enzyme is found to be far superior to these other options.

Strengths: It is a very interesting idea to do auto-diff at the LLVM level, and it is a very surprising insight that doing auto-diff after optimization results in a big performance advantage. Given the portability across languages that this approach provides, I could see it quickly becoming the favored way to perform auto-differentiation transparently, for most languages. I'm skeptical that this approach can compete in all cases with more conventional tools such as PyTorch, but it is very compelling to be able to auto-diff things like arbitrary libraries and then use in them in conjunction with more conventional tools such as PyTorch.

Weaknesses: The biggest weakness for me is that there is no discussion in the paper of the limitations of the approach. When can Enzyme do the wrong thing? I think it’s very important that the authors be up-front about this. Especially in low-level languages like C, programmers can (and do) write weird, obfuscated code that is going to be difficult/impossible to auto-diff (as an example, let’s say that a programmer uses bitwise operations to modify the exponent of FP number). Presumably there is no chance that Enzyme is going to be able to auto-diff such code. This seems to be the limitation of working at the LLVM level. If one works directly in C, it is possible to detect when the programmer is doing something dangerous that cannot be auto-diffed, and give a warning. But I would be concerned that Enzyme will happily auto-diff such code and come up with the wrong answer. Or, maybe I’m wrong and it is possible to precent this sort of problem, even at the LLVM level? Some sort of discussion of this is certainly warranted. I felt that while promising, one issue with the experiments is that it was quite unclear how the code to be auto-diffed was prepared. Adept is for C++, as far as I know, TAPENADE is for Fortran/C, and Enzyme is for any programming language that uses LLVM. Odient, for example (which is one of the sources for the benchmarks) is written in C++, so how was TAPENADE used? Overall, though, this is a strong paper.

Correctness: Yes, though as I wrote above, I would have really appreciated a careful discussion of failure modes.

Clarity: Yes, definitely.

Relation to Prior Work: Yes.

Reproducibility: Yes

Additional Feedback: After response and discussion: Looks like the paper will probably be accepted. Congratulations!!


Review 2

Summary and Contributions: This paper presents an automatic differentiation framework for LLVM that can be applied to multiple languages that use the LLVM compiler stack as well as applications that combine several languages. Because it runs after the LLVM IR is optimized, this compiler leads to better performance than some previous automatic differentiation systems that work at the source code level.

Strengths: This paper gets good performance results over past automatic differentiation systems that work at the source code level. Applying AD after standard compiler optimization passes also makes sense in many cases, although I imagine that it could sometimes harm performance. It's also clear that a lot of work has gone into this paper to integrate Enzyme into LLVM and support a wide range of use cases (e.g. both dynamically and statically compiled code).

Weaknesses: I wasn't sure how much is novel in the paper from a compilers and automatic differentiation perspective. Is this work just applying known AD techniques on a lower-level program representation (LLVM IR instead of a higher level language) or are there major new technical challenges that it has to solve to implement IR at this level? I imagine that some of the issues with type casts, allocating intermediate space, etc would also happen in C or C++ code and are not unique to LLVM IR. Unfortunately, the paper spends very little space talking about what's conceptually novel or difficult about implementing this in LLVM IR. I do understand that there can be performance benefits from doing optimization before applying AD though. My second concern is that the paper seems to primarily be written for a compilers/AD audience, and there is not much ML-specific content in it. It might be a better fit for a compilers conference than for NeurIPS overall, and it might also get more detailed reviews and more audience interest in those venues. The paper also has a few sections with very dense low-level details about LLVM, Julia, etc that are probably going to be very hard for anyone lacking experience with those systems to understand. Finally, the paper shows some great examples of where applying AD after optimizing the program helps improve performance, but I was wondering whether this can sometimes backfire: are there cases where the derivative of the optimized program is more expensive to compute than the derivative of the unoptimized program? I imagine that this might be true in some cases because compiler optimization passes are not designed to ensure that the derivative of the optimized code is also fast too compute.

Correctness: The paper seems technically correct. The evaluation covers seven benchmarks that span a variety of use cases, though it would have been nice to see more of them if there are other benchmarks that are used in recent AD papers (e.g. the Tapenade and Clad papers had some benchmarks that are not covered here, which would have offered more opportunity to compare performance against these systems).

Clarity: I found the paper pretty crunched for space in some places, although it was possible to understand the main ideas and experimental setup overall. I thought that the paper spent a bit too much space on the introduction and on some of the low level technical details, but too little space explaining which parts of the work are very different compared to past AD systems.

Relation to Prior Work: This is the first AD paper I've seen that operates at the compiler IR level, but it's not super clear whether this requires a significant advance in the algorithms used for AD or it is simply applying existing techniques on a different language. As I mentioned earlier, it seems that the same issues with type casts, undifferentiable instructions, shadow memory, etc would also occur if operating on C/C++ code. I am not an expert on AD but as a non-expert reviewer, I expected to see more details about what's novel there.

Reproducibility: Yes

Additional Feedback: I think that the paper in its current form may not be a great fit for NeurIPS, though I am okay accepting it if other reviewers find it exciting. I think that you can go in a couple of directions with this. If you are targeting an ML conference, consider writing more about what's new/interesting for a non-AD audience and showing more end-to-end application examples. Alternatively, a compilers/PL conference may be excited about this work as is if there are significant challenges in doing AD at the LLVM IR level that the past literature does not cover. Comments after author response: I'd like the authors to spend more time on the parts that are new for performing automatic differentiation on LLVM rather than other parts, but I'm okay accepting this paper if the other reviewers find it interesting.


Review 3

Summary and Contributions: The paper introduces Enzyme, an Automatic Differentiation system that works on low-level optimized LLVM IR. To enable this work, the paper had to build some interesting Type Analysis algorithms, to recover destroyed types and construct the correct backward formulas. Additionally, the system does additional but expected optimizations to compute gradients efficiently. It does pruning, inlining, caching and also using a compiler attribute (in the C frontend as illustrated) to allow opaque gradient functions for certain given functions. It handles static frontends such as C/C++ straightforwardly as expected. For dynamic frontends such as Julia, it adds additional infrastructure in the frontend binding to get runtime function definitions. Enzyme also provides convenient ffi interfaces for PyTorch and TensorFlow (and Julia). It uses a standard benchmark called ADBench from Microsoft showing convincing improvements over relevant competition. The authors also do an experiment ablating the gains on using post-optimization IR, and compare it to a reference implementation that runs the AD before optimization (i.e. an O2 pass of the compiler) is run. It is clear from the results that doing AD on the optimized IR provides significant gains. Overall, it's an impressive system.

Strengths: - fairly original work, has one close competitor (Clad) which is much weaker in many aspects - seems to be comprehensive, with Julia, PyTorch and TensorFlow interfaces, as well as completeness in the functionality of the autodiff - the empirical evaluation looks pretty good, with a good experimental setup

Weaknesses: - the project doesn't work on any accelerators, such as GPUs (this is a stretch to call a weakness, it's mostly out of scope), but other than that, I can't see any real weakness

Correctness: The claims are fairly correct. I cross-verified the prior work and what else is available in this space.

Clarity: The paper is fairly well written and easily understandable.

Relation to Prior Work: The paper clearly articulates differences to prior work, and is honest in it's claims. I am familiar with most of the prior work.

Reproducibility: Yes

Additional Feedback:


Review 4

Summary and Contributions: The authors propose Enzyme, an approach to take code for the forward pass, compile it into optimized LLVM code and then automatically construct its gradients using the intermediate LLVM representation. The authors claim that if the forward pass is itself used to generate the gradients then opportunities for optimization may be lost, thus resulting in suboptimal code. In experiments, Enzyme is compared to other approaches and shown to compare favorably resulting in faster runtimes across a variety of machine learning models. I have gone through the author response. I would encourage the authors to implement all the points they offer in their response.

Strengths: The introduction is well structured (although a bit unconvincing, see weaknesses). The authors do a reasonable job of introducing the problem using the example in Fig 1 which shows how opportunity to optimize code is lost if the forward pass is auto-diffed before. The experiments section also seem convincing. Section 2 delves into various aspects that show how much work it takes to come up with a system such as Enzyme.

Weaknesses: This is one of those papers which may be better evaluated if it were submitted to a PL conference or a systems conference. While the introduction reads well, I have to ask, the example in fig 1 seems like loop invariant code motion could have been performed on the source code itself. If so, do we really need to delve down to LLVM intermediate representation which introduces a host of complications (e.g., type analysis) ? Given that, fig 1 is the basis of the motivation, this example needs to be chosen carefully. I found it unconvincing. Perhaps adding another example where the optimization is simply not possible at source code level is warranted to nail down the introduction. Section 2 mentions a lot of aspects (type analysis, activity analysis, shadow memory etc.) but does not state which of these constitutes original research. For instance, in activity analysis, the authors state that this is something other systems already do. If so, what should I give this paper credit for? The experiments section is nice but I would have appreciated a few micro-benchmarks / case studies where Enzyme is shown to achieve optimized code for certain popular models that other approaches cannot achieve. Would have built up on the example in the intro to nail down the ideas underlying Enzyme and illustrated its virtues better, in my opinion.

Correctness: Empirical methodology seems correct. A few more case studies would have helped (see weaknesses). Claims seems correct but a bit unconvincing. Not quite convinced that we need to go down to LLVM IR to optimize model code. Why is it so difficult to convert forward pass source code to an intermediate representation that preserves all the goodness of the source code (e.g. types) while allowing auto-diff and optimization?

Clarity: The paper is well written. Some of my suggestions to improve the exposition may be found in weaknesses.

Relation to Prior Work: Yes

Reproducibility: Yes

Additional Feedback:

[Author Response · NeurIPS 2020]

We would like to thank the reviewers for their thoughtful advice and feedback. Reviewers raised concerns about the applicability of this work to NeurIPS. As AD is a fundamental computation required to train neural networks, perform Bayesian inference, and run many other ML algorithms, the AD tool-maker community has been well-represented at NeurIPS. In NeurIPS 2018, van Merrienboer et al survey AD and created a differentiable intermediate representation for arrays [3], de Avila Belbute-Peres et al published a differentiable physics engine [1], and in NeurIPS 2019 there was a dedicated workshop on Program Transformations for Machine Learning. This is also useful to the rest of the NeurIPS community as ML researchers rely heavily on AD tools like DSL's, PyTorch, and TensorFlow to enable their research.

Some reviewers were confused by the limitations of the approach. As we describe in lines 176-182, Enzyme is limited to producing gradients of programs whose active values have statically analyzable types and whose active functions all have statically available bitcode. Using bithacks to modify a floating-point value can result in values without statically analyzable types as it is ambiguous whether the value is a float or integer. Rather than produce incorrect code, Enzyme will emit a compile-time error if it is unable to perform an analysis needed by AD. This enables programmers to provide this information to the compiler in the form of additional attributes, a custom derivative, or other means.

Some reviewers had questions about what pieces of Enzyme's implementation are novel from other AD systems. The goal of Enzyme is to perform AD after optimization and as part of general-purpose compiler (LLVM), thereby enabling speedup over pre-optimization AD and AD on a variety of languages. This has historically not been explored as a result of the need to work on a lower-level, where information that existing tools rely upon has been lost. To remedy this, we introduce a new interprocedural type analysis (line 95) to derive type information and create a novel variant of shadow memory to support indirect function calls. Like most AD systems, we also implement well-studied (line 140) best practices such as activity analysis and caching (checkpointing), extending them to take advantage of the results of Type Analysis and LLVM's Alias Analysis for additional optimization.

A reviewer asked about how experiments were prepared. ADBench tests were directly taken from the benchmark suite, corrections and all. Additional tests were created using Tapenade's web interface or replacing programs with Adepts differentiable types (using Vector and Matrix extensions, where relevant). The two C++ tests (Euler and RK4) indeed aren't compatible with Tapenade and are shown with a red X in the plot and a N/A in the table, denoting benchmarks that don't work with that system (Figure 9).

A reviewer would have also liked to see Enzyme produce code for popular models. As part of our test suite we include a diverse set of ML techniques (LSTM, Gaussian Mixture Model, Bundle Analysis). The point of Enzyme, however, is not to replace frameworks like PyTorch or Tensorflow by better optimizing popular architectures, but to augment them by allowing ML researchers to use arbitrary existing code as part of their model without the effort of rewriting the codebase in that framework.

Some reviewers express concern that the example shown in Figure 1 of the submission could be remedied by rewriting the source code of the example. While identifying optimization opportunities is easy in small didactic examples, finding and performing such optimizations is tedious and non-obvious without automatic tools like compilers. For example, through the use of an automated tool, Doerfert et al found that "missing" function attributes in an already highly tuned DOE benchmark account for a 21% performance loss [2]. Doing so manually would require re-implementing techniques like alias analysis and code motion across thousands of lines of code. By performing AD alongside existing compiler analyses and optimizations, Enzyme can take advantage of all such optimizations automatically.

We thank the reviewers for discussing places of potential confusion. Upon further reflection, we also recognize that the paper's tone may have obfuscated certain points as well. We shall revise the writing to both clarify any confusion and generally be more objective. Finally, we would like to thank the reviewers for suggesting avenues of future research. Supporting AD of GPU and accelerator codes would definitely be valuable. While we don't explicitly address GPU's in this work, LLVM has both a frontend and backend for GPU's that should be able to integrate with Enzyme. While we've explored the specific ordering of optimizations in the experimental setup, fine-tuning the optimal place for AD in the stack remains a prime area for future work.

[1] Filipe de Avila Belbute-Peres, Kevin Smith, Kelsey Allen, Josh Tenenbaum, and J. Zico Kolter. End-to-end differentiable physics for learning and control. In S. Bengio, H. Wallach, H. Larochelle, K. Grauman, N. Cesa-Bianchi, and R. Garnett, editors, *Advances in Neural Information Processing Systems 31*, pages 7178–7189. Curran Associates, Inc., 2018.

[2] Johannes Doerfert, Brian Homerding, and Hal Finkel. Performance exploration through optimistic static program annotations. In *International Conference on High Performance Computing*, pages 247–268. Springer, 2019.

[3] Bart van Merrienboer, Olivier Breuleux, Arnaud Bergeron, and Pascal Lamblin. Automatic differentiation in ml: Where we are and where we should be going. In S. Bengio, H. Wallach, H. Larochelle, K. Grauman, N. Cesa-Bianchi, and R. Garnett, editors, *Advances in Neural Information Processing Systems 31*, pages 8757–8767. Curran Associates, Inc., 2018.


[Meta-Review · NeurIPS 2020]

All reviewers are positive about the paper. This paper is therefore accepted. Thanks for submitting to NeurIPS and please incorporate the reviewers' comment to further improve the camera ready version.